# OpenReview forum: "CARE-MI: Chinese Benchmark for Misinformation Evaluation in Maternity and Infant Care"
_NeurIPS.cc/2023/Track/Datasets_and_Benchmarks — NeurIPS 2023 Datasets and Benchmarks Poster_

### Official Review · Reviewer_dFCS · 2023-07-21
**cost-effective data construction approach for knowledge-intensive topics**

**Rating:** 6
**Confidence:** 3
**Correctness:** The dataset constructed in a sound way.

**Strengths:**

- The paper presents a novel and effective framework for constructing datasets tailored to knowledge-intensive topics, particularly in medical domains, which is potentially applicable to other datasets.

- The proposed dataset dealing with Chinese misinformation about maternity and infant care brings valuable contributions, especially for low-resource languages.

- The extensive experiments are conducted, including evaluation metrics such as correctness and interpretability. Both human evaluation and automated evaluation methods are utilized to ensure comprehensive evaluation.

**Additional Feedback:**

Please refer to the above “Opportunities for Improvement" section.

**Clarity:**

The paper is well-written and easy to follow.

(minor suggestions)
The captions and explanations about Figures and Tables need to avoid confusion. For example, (1) what does "supervised" mean in Table 1? (2) Also, what "N_E" indicates on the x-axis in Figure 1 in the Appendix? Is that the number of evidences?


**Documentation:**

Yes. The authors provide details about the proposed datasets and code.

**Ethics:**

No.

**Limitations:**

The authors discuss the limitations of the proposed dataset in Appendix C.

**Opportunities For Improvement:**

There are some points that need to be clarified regarding the dataset construction and evaluation details.

- Dataset

1. I would encourage the authors to provide more information (e.g., years of experience, degrees) about the experts who participated in the dataset construction and human evaluation. This could help validate the annotated results of the proposed dataset.

2. In Question generation task in Section 3, the authors mentioned that they carefully design and tune the prompt (in line 174). I suggest providing more details about the prompt engineering process, which could be helpful for applying the proposed framework in other domains.

3. During the "expert annotation" in Section 3, how did disagreements between two annotators occur? This information could help understand the difficulty of the annotation tasks.

4. The authors included samples with at least 3 supporting evidence. What is the motivation for choosing the number of evidence? Does the number of evidences affect the performance of the model?

5. I would suggest providing descriptive statistics of the proposed dataset, such as the average length, to help understand the characteristics of the proposed dataset.

- Benchmark and Evaluation

6. For evaluation, the 200 questions randomly selected from the benchmark for human baseline evaluation. How are the questions distributed across the four datasets (BIoS, CPubMed, MLEC-QA, MEDQA) in Table 4?

7. The LLMs were used with the prompts in Appendix B.5 for the evaluation. Were the prompts also tuned for better performance?

8. In Section 4.1, the authors recruited hired annotators to evaluate both correctness and interpretability. However, the results of the human evaluation of the interpretability score are missing.

9. In Section 4.3, the authors selected the answer outputs from MOSS-16B-SFT and ChatGLM-6B, which showed relatively lower performance in both correctness and interpretability scores. What motivated the selection of these two baseline models instead of the better performing ones?

**Relation To Prior Work:**

Yes. The paper clearly discusses the differences between this work and previous contributions, as shown in Table 1.

**Summary And Contributions:**

- This paper proposes a Chinese misinformation dataset related to maternity and infant care.
- The dataset comprises sets of (1) true/false question (or open-ended question), (2) evidences (relevant knowledge), and (3) correct and wrong answer with supporting paragraphs.
- This paper presents a cost-effective data construction approach for knowledge-intensive topics, utilizing LLMs with human annotations.

---

> ### Author Response · Authors · 2023-08-21
>
> **Regarding Opportunities For Improvement**
>
> Thank you for your comment.
>
> - **Dataset**
>
>     1. **Annotator Information.** The annotators for both data annotation and model evaluation are referred to the same group of people. All hired annotators are doctors (currently employed) with valid licenses and at least 10 years of experience. All annotators own at least bachelor's degree. We have provided more information regarding the corresponding part in the paper.
>
>     2. **Question generation details.** We obtain the final prompt for question-generation after the testing of different manually generated prompts.
>
>     3. **Inter-annotator agreement.** For the expert annotation in section 3, we calculate the Fleiss Kappa scores between the two annotators (not including the meta-annotator) for each guiding question that we ask the annotators to answer. In total we have four guiding questions, and the scores are presented in the table below.
>
>     4. **Choice of the number of evidence.** We choose this number as it ensures that each question is having relatively enough supportive evidences while the size of selected questions is not too small. The decision is made as this specific number seems to be the most reasonable one when doing the trade-off between the number of supportive evidences and the size of selected questions.
>
>     5. **Statistics of the benchmark.** We have added the corresponding statistics of the benchmark into the revised paper.
>
> | **Guiding questions**  | **Fleiss Kappa**                 |
> |------------------------|----------------------------------|
> | If only single answer  | 0.6565 (Substantial agreement) |
> | If question clear      | 0.5779 (Moderate agreement)      |
> | If answer correct      | 0.8180 (Almost perfect)          |
> | If evidence supportive | 0.7857 (Substantial agreement) |
>
> - **Benchmark and Evaluation**
>
>     1. **Distribution of selected samples for human evaluation.** They strictly follow their original distribution.
>
>     2. **Judgment model prompt.** The prompts for the judgment models are not tuned.
>
>     3. **Human evaluation results for interpretability.** We observe lower inter-agreement when evaluating the interpretability and thus the human evaluation on interpretability might be less convincing. Also, the task of evaluating interpretability is not the major focus of this paper and thus even though we would like to conduct such experiment, for budget reason we just cannot conduct the corresponding human evaluation.
>
>     4. **Choice of test set for judgment models.** To train the judgment models, it doesn't matter how the sentences have been generated. So we select answers that on average receive more borderline scores, i.e., closer to 0.5, which correspond to MOSS-16B-SFT and ChatGLM-6B. This choice ensures that the test set in general contains a more balanced distribution between good answers and bad answers.
>
> **Regarding Clarity**
>
> We have made modifications accordingly in the revised paper.

---

### Official Review · Reviewer_w6yk · 2023-07-21
**Review for CARE-MI: Chinese Benchmark for Misinformation Evaluation in Maternity and Infant Care**

**Rating:** 5
**Confidence:** 4

**Strengths:**

1. The authors provide a new type of dataset focusing on LLM performance in the particular subfield of maternity and infant care. The authors provide relatively extensive benchmarks of the performance of various LLM models on this dataset.

**Additional Feedback:**

None.

**Clarity:**

The paper could benefit from an improvement in writing.
1. References should all be cited at the end of the article and not in a mix of footnotes and references (websites can still be cited in the BibTeX form).
2. There are errors in spelling in the paper, including in Figure 1, where "Statements" is spelled "Satement". The authors should do a spelling check throughout the article.
3. The limitation section is included in the supplementary materials and easily missed. This should be rectified.

**Correctness:**

The reviewer is unable to verify the correctness of the dataset and the benchmarks due to lack of the dataset nor the benchmark code. The descriptions in the text appears to be adequate.

**Documentation:**

No, and this is included in the main comments.

**Ethics:**

No.

**Limitations:**

The limitation section should not be in the supplementary materials and should be in the main text as acknowledging limitations of benchmarks and the proposed dataset is a critical part of evaluation for the NeurIPS dataset track.

**Opportunities For Improvement:**

1. The reviewer was unable to locate the fine-tuned models trained by the authors nor the code to replicate the benchmarks. Although the authors state that this information would be released after the review period, given that the benchmark is the main contribution of this track at NeurIPS, it is critical to be able to evaluate the benchmark independently. This is a critical comment and one of the main bases for my rejection.
2. The reviewer is unsure the dataset adequately reflects the motivation of the paper. The authors state that the main motivation for this paper is to provide a dataset that could better understand if "expectant and first-time parents" should utilize LLMs for ,"e.g., pregnancy and/or baby-related illnesses, symptoms, milestones, etc". However, the source databases for the proposed benchmarks come from abstracts and central articles from PubMed (BIOS, CPubMed) and the National medical Licensing Examination in China (NMLEC), both of which are formal databases using language in the written style. The provided dataset reflects such tendency where each question contains significant medical jargon that expectant and first-time parents cannot reasonably be expected to have, such as "IgG“ and "Bilirubin". The reviewer believes it is dangerous to _assume_ that the performance of LLM on formal medical documents would actually translate to questions asked by expectant parents, who likely have much less medical knowledge, would use significantly more descriptive language rather than medical lingo, and in general would be vaguer in the questions that they are trying to ask. If the goal is to see if LLMs would provide misinformation for new parents, the reviewer is unconvinced the current dataset and benchmarks achieve this. I think the authors would benefit from a rethink on what this dataset actually aims to achieve. The authors acknowledge this in the limitation section but the reviewer believes this acknowledgment is insufficient in addressing this concern. This is a critical comment and one of the main bases for my rejection.
3. The dataset is relatively small and in the view of the reviewer are missing many common scenarios that one would expect new parents to ask while including many scenarios that would only reasonably be useful to medical doctors in the OBGYN field. FOr example, one question is: "what is the drug of choice for Convulsions in neonates with hypoxic-ischemic encephalopathy"?

**Relation To Prior Work:**

The authors have adequately discussed prior work.

**Summary And Contributions:**

The authors present CARE-MI, a new benchmark designed for evaluating LLM misinformation in a sensitive domain (maternity and infant care) and a language other than English (Chinese). This is one of the first attempts to explore misinformation evaluation in non-English LLMs and specific knowledge-intensive topics.  The CARE-MI benchmark consists of 1,612 expert-filtered questions along with human-selected references, and extensive experiments were conducted using the benchmark, revealing that existing Chinese LLMs perform sub-optimally in the context of maternity and infant care.

---

> ### Author Response · Authors · 2023-08-21
>
> **Regarding Opportunities For Improvement**:
>
> We thank the reviewer for the comments. The reviewer is unable to find the model weights for the fine-tuned judgment models and also cannot locate the corresponding codes for reproducing the benchmark. Furthermore, the reviewer is also concerned about the misalignment between the constructed benchmark and the target that we aim to benefit. We appreciate the reviewer for the feedback.
>
> 1. **Model weights.**
>
>     - We have released the model weights for the LLaMA-13B-T which we fine-tune and utilize as judgment models (please see our comment to all reviewer). The details of how the models are fine-tuned were described in the original paper, and we will add a script that can be used to fine-tune the judgment models to our code repository. The link to our [Github code repository](https://github.com/Meetyou-AI-Lab/CARE-MI/) can be found in the supplementary materials (Appendix), where we describe how we construct the benchmark.
>
>     - Please kindly check the supplementary materials which can also be accessed through OpenReview. The code to replicate the benchmarks is presented in the [Github code repository](https://github.com/Meetyou-AI-Lab/CARE-MI/) that we have listed in the supplementary materials (Appendix).
>
> 2. **Dataset and motivation.** We agree with the points mentioned by the reviewer, yet we recognize that the critique may stem from the ambiguity in our paper. The sentences that the reviewer has raised concerns about are located from line 64 to 68. The point that we initially tried to make is that the domain of maternity and infant care is of vital importance and needs special attention. We refer to "expectant and first-time parents" solely to emphasize the significance of the corresponding domain. This importance motivates us to build the benchmark for this domain so that the misinformation of LLMs in this domain can be quantified using the benchmark and eventually people like expectant and first-time parents might benefit from it. We are aware of the gap; However, even though not originally designed for user-LLM interaction cases, our benchmark is still beneficial for indicating LLMs' performance as we think that LLM performing well in our benchmark serves as a necessary but not sufficient condition for doing well in those cases. We consider the task of resolving real-world user queries in the maternity and infant care domain as the natural next step for this work; in our current work, the models are only required to answer questions with standards norms and specialized knowledge; user queries tend to be more ambiguous and complex, potentially necessitating enhanced reasoning capabilities from the models for concrete resolution. We have modified the corresponding sentences to better present the idea that we are trying to convey; this modification is reflected in the revised paper.
>
> 3. **Dataset size.** We acknowledge that our proposed benchmark has a relatively small size in comparison with others. However, we do think that dataset being small in this domain is natural and reasonable. For a knowledge-intensive domain like maternity and infant care, building a dataset with expert-level supervision is complicated and requires additional effort. Also, similar datasets also have limited sizes, e.g., TruthfulQA with only 817 samples. Furthermore, our proposed framework allows us to produce samples at scale with better efficiency and thus more samples can be relatively easily included into the benchmark if provided with more valid data sources. As for the point of not being able to cover common cases asked by new parents, we refer the reviewer to our answer to the previous point. This modification is reflected in the revised paper.
>
> **Regarding Limitations**
>
> We move the limitation section to the main text accordingly. This is presented in the revised paper.
>
> **Regarding Clarity**
>
> 1. We have made modifications accordingly in the revised paper.
> 2. We have done another round of proofreading and made modifications accordingly.
> 3. We have moved it into the main paper.
>
> Again, we thank the reviewer for providing feedback.

---

> > ### Comment · Reviewer_w6yk · 2023-08-28
> >
> > I appreciate the authors' work to address my comments. I am still of the view that the dataset and the motivation mismatch is a real pain point of this dataset and remains a major concern but I have updated my score to reflect the fact that the reproduction dataset and code has been released.

---

### Official Review · Reviewer_aQxk · 2023-07-24
**A well-written paper and a dataset on an important problem**

**Rating:** 7
**Confidence:** 4
**Correctness:** I did not find any major flaws in the…
**Clarity:** The paper was very well written and e…

**Strengths:**

* The paper provides a very useful dataset, carefully constructed. It could help document and quantify gaps in misinformation in LLMs
* It combines knowledge sources from various sources to verify the validity of the labels in the data, and further filtered by subject matter experts
* Paper is fairly well written and was easy to follow.

**Additional Feedback:**

N/A

**Documentation:**

The dataset description in the paper was fairly comprehensible. The appendix gives more details. But I did not find any datacard or data statement.

**Ethics:**

No major ethics issues as this is building an evaluation resource using machine generated text, to evaluate gaps in model capabilities. However, I do urge the authors to add a section outlining the limitations of the work, as well as broader impacts.

**Limitations:**

* Like any work in this space, this paper/dataset is also limited to a specific domain and the insights from this study may not generalize to other domains. But that is not necessarily a limitation, but it would be good to highlight this as a limitation, further. Currently the manuscript does not contain a limitations section or an ethical considerations section. I strongly suggest the authors to include it.

**Opportunities For Improvement:**

* I think the introduction could be framed a bit better by first establishing the major gaps that this paper is fixing. The intro currently lays out the challenges/issues at large in LLM space, and the misinformation issue, but it could do a better job of highlighting the importance of medical misinformation, and especially maternity and infant care, and how new parents around the world rely on online advices for this. Currently the domain is just introduced without much motivation. Similarly, you could better motivate the need for doing such work in non-English languages (demonstrating the Western focus in existing literature in this space).

**Relation To Prior Work:**

The paper seems to have situated the work fairly well in existing literature. I would have liked a bit more discussion from social sciences focused on technology based interventions in healthcare or public health in general. While this is a NeurIPS paper, I think it would be good to situate the work in that space too.

**Summary And Contributions:**

This paper provides a dataset to evaluate LLM generated misinformation in long form generations in Chinese, in the topic of  maternity and infant health. As we increasingly rely on online sources for various healthcare related questions, and as LLM based generated content is increasingly becoming commonplace, this dataset and paper will help to bring attention to the misinformation issue, as well as quantify various approaches to mitigate it.

---

> ### Author Response · Authors · 2023-08-21
>
> **Regarding Opportunities For Improvement**
>
> Thank you for the advice. We have made modifications to better emphasize our focus in the revised paper.
>
> **Regarding Limitation**
>
> Thank you for your advice. We actually have already included a limitation section in the appendix, inside the supplementary materials. Since it is easy for readers to miss it, we have moved it to the main paper.

---

> > ### Comment · Reviewer_aQxk · 2023-08-28
> >
> > Thanks for your edits.

---

### Official Review · Reviewer_5eH2 · 2023-07-24
**A Chinese Benchmark for Maternity and Infant Care**

**Rating:** 6
**Confidence:** 3
**Correctness:** See Above.

**Strengths:**

1. There only exists limited datasets in the domain of maternity and infant care domain (e.g. MATINF). The proposed dataset could be beneficial to develop LLM applications in this specific domain.
2. The proposed dataset construction pipeline is straightforward and easy to follow.
3. Documentation of the proposed dataset (the GitHub repo) is clear and well-organized.

**Additional Feedback:**

No

**Clarity:**

In general, the presentation of the paper is clear as the whole idea of the paper is straightforward.

**Documentation:**

This paper missed several details on data collection, to name a few:
1. For true statement generation: rule-based methods are widely used to generate true statements from KG/MCQ datasets. However, the paper did not mention how exactly these rule-based methods are performed. Only the prompt used for generating declarative statements based on GPT-3.5-turbo is provided in Appendix.
2. For false statement generation: similarly, the paper claimed that "construction is also done by combining a rule-based method with applying the GPT-3.5-turbo". I am not sure how to combine the rule-based method with GPT.
3. For the evaluation of ChatGPT, ChatGLM, and ChatYuan for the question generation task, accuracy is used to compare these three models, which is quite confusing. How exactly did you evaluate the quality of the generated questions? Did you perform a human evaluation? Why do you use accuracy rather than other automatic natural language generation metrics?
4. For knowledge retrieval, line 189 claimed that "retrieve the three most relevant knowledge". How to define "knowledge" here? Does it mean a sentence or paragraph? It is quite confusing.
5. How to define "long-form" generation? What is the average length of answers for questions from different sources?
6. How about the inter-annotator agreement? What are the demographics of the annotators? Are they doctors or graduate students in the medical school? Are they the same in data annotation and model evaluation?

**Limitations:**

As shown above.

**Opportunities For Improvement:**

1. This paper claimed that their dataset is "for Misinformation Evaluation in Maternity and Infant Care". It seems to me that the proposed dataset is just a question-answering dataset that only has questions and retrieved evidence by using BM25 in the maternity and infant care domain. I am not sure if a model that cannot answer the question correctly should be viewed as creating misinformation.  As shown in Table 1, this paper seems to consider all question-answering datasets as misinformation evaluations. What is the relation between this work and other efforts on misinformation and disinformation, such as fact-checking, rumor detection, etc?
2. The "long form generation" concept emphasized by this paper is underdefined. How long an answer can be considered a "long form" answer? In the annotation guideline, the first one is "the sample question has at most one single correct answer". The proposed task is not a classification task. How can you guarantee there only exists one long-form answer?
3. The decoding strategy for all the baselines is not mentioned in the paper. For example, did you try different prompts/instructions, chain-of-thoughts,  or exemplars (in context learning) in the inputs? What are the hyper-parameters such as temperature, and the maximum length used for decoding? It seems to me that these strategies affect the performance of the baselines a lot.
4. I have concerns about the retrieved evidence for the question. The knowledge retrieval is simply based on BM25 and the only control is in the expert annotation stage (if the annotator thinks it contains enough knowledge). How the retrieved knowledge affects the questions answered is unknown.
5. As shown in Table 4, the evaluation of the proposed benchmark has a very high variance (around 20%-30%). It is understandable as all the evaluation is based on two annotators. However, such a setting makes the results hard to replicate and compare with other LLMs.
6. I am not sure of the benefits of constructing the datasets from the four mentioned datasets in the paper (MLEC-QA, BIOS, CPubMed-KG, MEDQA) and the proposed pipeline in the paper. It seems to me that you can get the true statement from the articles from Wikipedia/PubMed, then mutate them into false statements/questions. Related evidence can be obtained from the relevant paragraphs in the corresponding articles, similar to the knowledge-intensive tasks in KILT [1], such as FEVER [2].

References:
[1]KILT: a Benchmark for Knowledge Intensive Language Tasks. NAACL 2021
[2]FEVER: a large-scale dataset for Fact Extraction and VERification. NAACL 2018

**Relation To Prior Work:**

1. The paper only mentioned the MATINF in lines 71-72 ("does not contain expert-level annotations on the samples") but did not discuss how this work differs from it in detail.
2. The paper claims that Table 1 summarizes previous datasets in misinformation evaluation, which is confusing. Most of the datasets listed in Table 1 are just question-answering datasets. It seems to me that only TruthfulQA is related to misinformation to some extent. I do not why other QA datasets are viewed as evaluating misinformation.

**Summary And Contributions:**

This paper proposes a Chinese benchmark for misinformation evaluation in long-form generation for large language models (LLMs) on the topic of maternity and infant (care CARE-MI), which contains 1,612 expert-filtered questions, accompanied by human-selected references. A pipeline consists of five stages: true statement generation, false statement generation, question generation, knowledge retrieval, and human annotation is proposed to generate the synthetic data. GPT4, GPT3.5-turbo, and a couple of Chinese LLMs including MOSS, ChatGLM, and BELLE are evaluated on the proposed benchmark.

---

> ### Author Response · Authors · 2023-08-21
>
> **Regarding Opportunities For Improvement**
>
> We thank the reviewer for the comments.
>
> 1. **Difference between our work and previous work.** Sorry that we are not explicitly explaining some important concepts in the paper. We follow existing definition of misinformation from Weidinger et al. [1] where the misinformation is defined as a harm that "rise from the LM outputting false, misleading, nonsensical or poor quality information, without malicious intent of the user". A key distinction between their defined misinformation and other types of harm incited by false information is that in misinformation the false information is not generated deliberately but naturally and unintentionally, whereas other types of harm might stem from the malicious intents of users. In our case, we consider a model unintentionally generating factually incorrect answers as creating misinformation. As also has been mentioned in [1], such misinformation might even cause physical harm and other unpredictable consequences, especially in knowledge-intensive domains such as medical and law, in contrast to general-domain/open-domain. This motivates us to build our benchmark, which focuses on a knowledge-intensive sub-domain, maternity and infant care. In addition, we summarize question-answering datasets that focus on knowledge-intensive domains in Table 1. Here we list a few differences between us and the other related work:
>
>     - Our work focuses on detecting the misinformation problem following the definition in [1], especially in the knowledge-intensive domains instead of general-domain/open-domain. This is different from KILT [2] and FEVER [3] where the authors construct tasks given general domain Wikipedia text. We want to highlight that constructing datasets for specific knowledge-intensive domains instead of general-domain is more complex and requires additional efforts and thus we do believe that our work, which includes building a benchmark and proposing a transferable framework, is not trivial.
>
>     - We frame the task in the format of question-answering instead of others for reasons. We intentionally choose this specific format of evaluation since we are more concerned about the misinformation produced during LLMs' generation, especially when they are generating long free-form text. This is different from some previous work where they apply multiple-choice question-answering to evaluate misinformation in LLMs; we believe that this method cannot fully evaluate the misinformation issue which can be triggered more frequently during the generation of relatively long sentences.
>
>     - According to the definition of misinformation in [1], the harm is generated inherently and unintentionally by the LLMs without leveraging any malicious intents from users. This is different from rumor detection where the false information is usually generated intentionally. Besides, since we are evaluating LLMs' output given questions, the answers from the models not only need to be factually correct but also need to be relevant to the question and logically reasonable. This makes our task different from fact-checking, which only cares about whether the given fact is true or not.
>
>       We have modified the paper accordingly; the changes are reflected in the revised paper.
>
> 2. We modify the paper and add more clarification regarding this your comments.
>
>     - **Definition of long-form generation.** Regarding the first question in the comments, "long-form generation" is a widely used concept that can be found in question-answering papers [4] and text generation papers [5,6], usually referred as generated natural language that "span multiple sentences" [4] or "paragraph-length" [6]. We found no stringent standard that can be utilized to distinguish long-form generation against shorter ones; however, the way that we evaluate LLMs put nearly zero limitation on the model generation, i.e., we let the models generate sentences as long as they prefer unless the maximum length is reached. In contrast to evaluations using multiple-choice question-answering where the models are only required to generate a few tokens, the generated answers in our cases contain 4.5 sentences on average and the average number of tokens for each answer is 123.1.
>
>     - **Guarantee for having a single long-form answer for each question.** Regarding the second question in the comments, by saying "the sample question has at most one single correct answer", what we really mean is that even though an answer to the question can have various superficial forms, the facts and information behind all true answers should be the same. So we are not actually restricting the answers by their forms, but the content that they contain.

---

> > ### Author Response · Authors · 2023-08-21
> >
> > **Regarding Opportunities For Improvement (Continued)**
> >
> > 3. **Hyper-parameter settings.** We forget to mention the actual decoding method that we utilize in model generation. For all the models evaluated, we apply greedy decoding during their generation. Comparing different decoding strategies is out of the scope of this paper and thus we simply apply greedy decoding strategy to all generation cases as it is considered to be the most standard way for text generation. As we have claimed in the paper, we do not apply any prompt during the question-answering and all models are evaluated under complete zero-shot settings, i.e., for each question, the input of the model is the question itself. We mention in the supplementary materials that we use a temperature of 0 for both models from the GPT family and 0.01 for the rest models for better output consistency; the maximum length during decoding for all models is set to 2,048. We have modified the paper accordingly to better present the experiment details.
> >
> > 4. **Influence of the retrieved knowledge.** The evidence is not utilized in the model evaluation as the model evaluation is done in complete zero-shot settings. We add an ablation study to the revised paper to show that the retrieved knowledge is useful for improving the performance of judgment models. The comparison results are shown in the below table. From the following table we can see that adding evidence to the fine-tuning samples significantly increases the  performance of judgment models, especially for correctness.
> >
> > |                      | **Without Evidence** | **Without Evidence**  | **With Evidence** | **With Evidence**|
> > |----------------------|---------|----------|----------|----------|
> > |                      | Pearson | Accuracy | Pearson  | Accuracy |
> > | **Correctness**      | 0.779   | 0.806    | 0.868    | 0.898    |
> > | **Interpretability** | 0.639   | 0.867    | 0.683    | 0.835    |
> >
> > 5. **High variance in Table 1.** The variance is not calculated between the annotators but among all evaluated results and thus the variance is not reflecting the disagreement between the annotators but the distribution of the evaluated scores. For each question, we take the average between the three annotators (please see our message to all reviewers for updated information regarding this part) and present the average scores over all questions for each model in Table 1. We additionally calculate the inter-annotator agreement between the two annotators; the Fleiss Kappa agreement score for correctness is 0.7547 (substantial agreement) and the score for interpretability is 0.5728 (moderate agreement). Note that we hire a third expert as the meta-annotator to resolve each disagreement.
> >
> > 6. **Benefits of our Benchmark.** Wikipedia only covers general-domain knowledge while generating questions from PubMed requires expert-level annotations and thus not ideal for efficiently producing samples at scale. Our proposed framework aims at reducing the cost when building question-answering datasets in a totally new knowledge-intensive domain by constructing the samples on top of existing work. We respect and leverage existing efforts to avoid building the benchmark from scratch; beyond better efficiency, leveraging current work also allows us to take the advantage of some specially designed datasets, such as MLECQA, which includes many scenario-based questions asking about very specific symptoms, that can be hardly found in either Wikipedia and PubMed. Nonetheless, we add a few sentences to clarify the corresponding concerns accordingly.
> >
> > **Reference**
> >
> > [1] Weidinger, Laura, et al. "Taxonomy of risks posed by language models." Proceedings of the 2022 ACM Conference on Fairness, Accountability, and Transparency. 2022.
> >
> > [2] Petroni, Fabio, et al. "KILT: a benchmark for knowledge intensive language tasks." arXiv preprint arXiv:2009.02252 (2020).
> >
> > [3] Thorne, James, et al. "FEVER: a large-scale dataset for fact extraction and VERification." arXiv preprint arXiv:1803.05355 (2018).
> >
> > [4] Fan, Angela, et al. "ELI5: Long form question answering." arXiv preprint arXiv:1907.09190 (2019).
> >
> > [5] Köksal, Abdullatif, et al. "Longform: Optimizing instruction tuning for long text generation with corpus extraction." arXiv preprint arXiv:2304.08460 (2023).
> >
> > [6] Cho, Woon Sang, et al. "Towards coherent and cohesive long-form text generation." arXiv preprint arXiv:1811.00511 (2018).

---

> > > ### Author Response · Authors · 2023-08-21
> > >
> > > **Regarding Relation To Prior Work**
> > >
> > > 1. **Comparison with MATINF.** MATINF contains three parts, and we compare our benchmark with the QA part of MATINF. Both our benchmark and MATINF contain QA pairs where both questions and answers are written in natural language. MATINF contains no supporting documents; our benchmark contains QA pairs and additional evidences checked by human experts. MATINF contains QA pairs collected from only forum where both questions and answers are written by non-experts; our benchmark contains QA pairs either constructed or collected from authoritative data sources like NMLEC, and each question is reviewed by experts. Text in MATINF are vaguer while text in our benchmark in general contains more medical terms and have more precise description due to their difference in source of collection. MATINF is not designed for general QA task while our benchmark is designed to test the misinformation level of LLMs and is equipped with judgment models for better evaluation. We have added a few sentences to the revised paper to highlight the comparison accordingly.
> > >
> > > 2. **QA datasets in Table 1.** The reason that we think the datasets listed in Table 1 can be utilized for misinformation evaluation is that these datasets can be used to test if the model is outputting factually incorrect information unintentionally. TruthfulQA is initially designed to evaluate the imitative falsehoods of the LLMs, and is naturally suitable for misinformation evaluation. Other datasets mentioned in Table 1 can also be utilized for misinformation evaluation even though they might not be initially designed for this purpose.
> > >
> > > **Regarding Documentation**
> > >
> > > Thank you for the comments.
> > >
> > > 1. **True statement generation.** This is actually presented explicitly in the code repository. We have added a few sentences to the appendix in the revised paper to present how we combine the two systems together.
> > >
> > > 2. **False statement generation.** This is also presented explicitly in the code repository. We have added a few sentences to the appendix in the revised paper to present how we combine the two systems together.
> > >
> > > 3. **Evaluation of QG quality.** We actually have mentioned how we evaluate the question-generation quality in the supplementary materials (Appendix). Specifically, we evaluated the quality of the 100 generated questions through human evaluation. Given the importance of question generation, we opted for more stringent evaluation criteria, as opposed to relying solely on natural language generation metrics. We adopted three evaluation criteria which were mentioned in Appendix B.3: 1) fluency, the sentence must be fluent in the Chinese context (eg. word order and sentence structure), 2) consistency, where the generated questions should ask about the same entities or events mentioned in the true statements, and 3) Directness, that a legal question should not directly contain the correct answer. If a generated question fails any of the above rules, then it is considered incorrect. Ultimately, the ChatYuan performed the best and was used to generate both True/False (TF) and open-ended (OE) questions.
> > >
> > > 4. **Definition of knowledge.** Sorry for the confusion. Here each knowledge is referred as a single paragraph, either from the textbook or from Wikipedia pages.
> > >
> > > 5. We answer the questions in separate bullet points.
> > >
> > >     - **Definition of long-form generation.** Please see our previous answer regarding long-form generation.
> > >
> > >     - **Average length of answers from different sources.** Please the table below for details.
> > >
> > > |  **Source** | **Avg. #Sentence** | **Avg. #Token** |
> > > |---------:|:-------------------:|:---------------:|
> > > | bios    | 4.4286             | 104.8442        |
> > > | cpubmed | 4.7790             | 129.4619        |
> > > | medqa   | 4.5621             | 129.6878        |
> > > | mlecqa  | 4.5392             | 128.4182        |
> > >
> > > 6. **Inter-annotator agreement.** We hire three experts (two for annotation and one as the meta-annotator to resolve disagreement) for the question filtering (Section 3) and three experts (all of them for annotation) for the LLM evaluation (Section 4). For question filtering, the annotation agreement is presented in the table below. For LLM evaluation, the Fleiss Kappa agreement score for correctness is 0.7547 (substantial agreement) and the score for interpretability is 0.5728 (moderate agreement). All hired annotators are Chinese. All hired annotators are doctors (currently employed) with valid licenses and at least 10 years of experience. The annotators for both data annotation and model evaluation are referred to the same group of people.

---

> > > ### Comment · Reviewer_5eH2 · 2023-08-23
> > >
> > > Thanks for the clarification. The authors provide more results to justify their statements and address many of my concerns. I appreciate the efforts. I would like to increase my rating to 6.

---

### Author Response · Authors · 2023-06-14
**Link to the proposed benchmark dataset**

Dear reviewers, area chairs, and program chairs:

The link to the proposed benchmark dataset is attached here: [https://drive.google.com/drive/folders/18nRlBfwoHvi1EwEN_EcNH5GZPAAZOciW?usp=sharing](https://drive.google.com/drive/folders/18nRlBfwoHvi1EwEN_EcNH5GZPAAZOciW?usp=sharing).

Specifically, the proposed benchmark is contained in the **care-mi.tsv** file which you can access through the above link.

Kind regards,\
Tong Xiang

---

### Author Response · Authors · 2023-08-21
**To all reviewers**

We would like to thank all the reviewers for the time and efforts made in providing comments and feedback.

1. We would like to correct and update one misstatement in the paper:

    - In our original submitted paper, we state that "For each model-generated answer, we recruit two expert annotators...". This statement is not correct as we actually hire three annotators for this evaluation and take the average scores over the three annotators for each answer from each evaluated model.

2. **We have submitted the revised paper, along with the supplementary materials, in the same pdf. Newly added sentences and tables are highlighted in pink.**

3. We will have the [code repository](https://github.com/Meetyou-AI-Lab/CARE-MI) updated in a few days.

4. The judgment models for evaluating correctness and interpretability can now be accessed through huggingface; we will have the instruction regarding how to use them updated in the code repository in the next few days:

    - [Correctness](https://huggingface.co/tx39/llama-13b-T-caremi-judgment-correctness)
    - [Interpretability](https://huggingface.co/tx39/llama-13b-T-caremi-judgment-interpretability)

---

### Decision · Program_Chairs · 2023-09-22

**Decision:**

Accept (Poster)

**Comment:**

This paper introduces a new Chinese benchmark, CARE-MI, designed for evaluating LLM misinformation in the maternity and infant care subfields. The dataset includes 1,612 questions, answers and supporting evidence.
The authors also present a cost-effective data construction approach for knowledge-intensive topics and a comprehensive evaluation using manual and automatic metrics.

The paper is well-written. This new benchmark will be useful to evaluate misinformation in Chinese LLMs.
The reviewers had several questions related to the dataset characteristics and creation method. The paper would benefit from including these answers/explanations.